# Characteristics of Serum Exosomes after Burn Injury and Dermal Fibroblast Regulation by Exosomes *In Vitro*

**DOI:** 10.3390/cells12131738

**Published:** 2023-06-28

**Authors:** Jie Ding, Yingying Pan, Shammy Raj, Lindy Schaffrick, Jolene Wong, Antoinette Nguyen, Sharada Manchikanti, Larry Unsworth, Peter Kwan, Edward Tredget

**Affiliations:** 1Wound Healing Research Group, Department of Surgery, Faculty of Medicine and Dentistry, University of Alberta, Edmonton, AB T6G 2S2, Canada; pan16@ualberta.ca (Y.P.); lschaffr@ualberta.ca (L.S.); jolene.wong@ualberta.ca (J.W.); atn2@ualberta.ca (A.N.); sharada.manchikanti@albertahealthservices.ca (S.M.); pkwan1@ualberta.ca (P.K.); etredget@ualberta.ca (E.T.); 2Department of Chemical and Materials Engineering, Faculty of Engineering, University of Alberta, Edmonton, AB T6G 2S2, Canada; shammy@ualberta.ca (S.R.); lunswort@ualberta.ca (L.U.)

**Keywords:** exosomes (EXOs), burns, wound healing, hypertrophic scarring, cytokines

## Abstract

(1) Background: Exosomes (EXOs) have been considered a new target thought to be involved in and treat wound healing. More research is needed to fully understand EXO characteristics and the mechanisms of EXO-mediated wound healing, especially wound healing after burn injury. (2) Methods: All EXOs were isolated from 85 serum samples of 29 burn patients and 13 healthy individuals. We characterized the EXOs for morphology and density, serum concentration, protein level, marker expression, size distribution, and cytokine content. After a confirmation of EXO uptake by dermal fibroblasts, we also explored the functional regulation of primary human normal skin and hypertrophic scar fibroblast cell lines by the EXOs *in vitro*, including cell proliferation and apoptosis. (3) Results: EXOs dynamically changed their morphology, density, size, and cytokine level during wound healing in burn patients, which were correlated with burn severity and the stages of wound healing. EXOs both from burn patients and healthy individuals stimulated dermal fibroblast proliferation and apoptosis. (4) Conclusions: EXO features may be important signals that influence wound healing after burn injury; however, to understand the mechanisms by which EXOs regulates the fibroblasts in healing wounds, further studies will be required.

## 1. Introduction

A burn is a severe skin injury that leads to a complex and dynamic response. The response involves multiple organs beyond the integumentary, including the nervous, endocrine, and hematopoietic systems through the signaling of numerous types of cells by a wide range of mediators and leads to overlapping stages of wound healing, including hemostasis, inflammation, proliferation, and remodeling [1,2]. Clinically, the duration and intensity of each stage depend on the severity of the injury, its location on the body, and the patient’s age, race, and underlying health. These conditions can lead to normal repair or pathologic scar formation such as hypertrophic scars (HTS). The mediators include cytokines released after burns that may play a decisive role in signaling local cells and guiding wound healing [3].

Exosomes (EXOs) are endosome-derived extracellular vehicles (EVs) secreted by all cells of prokaryotes and eukaryotes and are thought necessary for intercellular communication by transferring specific mechanism-derived accumulated cellular components to recipient cells [4]. Studies have reported that EXOs derived from stem cells, immune cells, and fibroblasts play a crucial role in various stages of wound healing, including inflammation, proliferation, and tissue remodeling [5,6]. In recent years, the diagnostic and therapeutic potential and advantages of EXOs over traditional protocols have been widely publicized in the field of cutaneous wound healing [7]. However, more research is needed to fully understand EXO characteristics and the mechanisms of EXO-mediated wound healing, especially wound healing after a burn injury.

We hypothesize that in responding to burn injury, EXOs are released into the blood and carried by the circulation to the wound site, where they guide local dermal cell behavior by delivering mediators such as cytokines. The severity of the injury determines the function of EXOs, which would direct normal tissue repair or pathological wound healing such as hypertrophic scarring. In this study, we characterized the features of serum EXOs such as morphology, density, size distribution, concentration, protein concentration of lysed EXOs, marker expression, and determined cytokines inside the EXOs. We also explored functional regulation by the EXOs in primary human dermal fibroblasts *in vitro*, including cell proliferation and apoptosis. We believe that this investigation will assist in understanding the natural behavior of serum EXOs in burn patients and the mechanism whereby EXOs participate in wound healing as well as scar formation.

## 2. Materials and Methods

### 2.1. Patients and Blood Samples

Eighty-five peripheral blood samples were collected from 29 adult burn patients, 23 male (M) and 6 female (F) with a mean age 45.0 ± 15.5 years (y) and a range of severity based on total burn surface areas (TBSA) at multiple time points after the initial burn injury. Thirteen healthy individuals, 6M/7F of age 36.1 ± 16.1 y were used as controls, whose blood samples were collected once. All samples were obtained with written informed consent under a protocol approved by the University of Alberta Hospital Health Research Ethics Board.

### 2.2. EXO Isolation

All the EXOs were isolated from the serum, utilizing a total EXO isolation reagent (4478360, ThermoFisher Scientific, Rockford, IL, USA) based on the manufacturer’s instructions. The blood samples collected from the burn patients and controls were left to clot at room temperature for 30 min. After the clot was removed, the serum was harvested by centrifuging each sample at 2000× *g* for 10 min. Cells and debris were removed by centrifuging the serum samples at 2000× *g* for 30 min. The EXO isolation reagent was added to the serum at a ratio of 1:5 (reagent: serum) and mixed well by a vortex until the solution was homogenous. Each sample was incubated at 4 °C for 30 min. The EXOs were collected by centrifuging each sample at 10,000× *g* for 10 min at room temperature. The EXO pellets were kept at −20 °C until further experimentation.

### 2.3. Transmission Electron Microscope (TEM)

Twenty EXO pellets taken on at least three occasions after the initial injury were taken from 6 burn patients (4M/2F, 48.0 ± 7.3 y) with various TBSA, along with 4 pellets from 4 controls (3M/1F, 30.9 ± 14.3 y). The pellets were suspended with PBS at a ratio of 1: 0.5 (serum: PBS; *v*/*v*) and fixed with 1% paraformaldehyde (sample: 1% PFA = 10:1) for 30 min. The fixed EXO samples were diluted 1:10 with 0.1 M hepes buffer solution. The diluted samples (10 μL each) were loaded on a parafilm. A formvar/carbon film grid was coated on the top of the sample and incubated for 7 min. After 3 washes with ddH_2_O for 2 min each, the samples were stained with 2% uranyl acetate in H_2_O for 7 min. Excess uranyl acetate solution was blotted off with filter paper, and the samples were air-dried overnight. EXOs were imaged using a JEM-2100 TEM (JEOL Canada, Inc., St-Hubert, QC, Canada) at 100 K and 400 K magnification.

### 2.4. Western Blot

EXO pellets from two burn patients and one control were lysed with RIPA lysis buffer (20-188, MilliporeSigma, Burlington, MA, USA). The supernatant was collected after centrifuging the pellets at 12,000× *g* for 10 min. Cell lysates were prepared from HTS fibroblasts using the same lysis buffer for calnexin expression as a positive control. The protein concentration was determined using BCA protein assay (#23227, ThermoFisher Scientific, Waltham, MA, USA) by a VARIOSKAN LUX microplate reader (ThermoFisher Scientific, Waltham, MA, USA). Albumin standards were purchased from ThermoFisher Scientific (#23209, Waltham, MA, USA).

Protein samples (10 µg each) were resolved by electrophoresis on 10% Mini-PROTEAN TGX gel (Bio-Rad Laboratories, Hercules, CA, USA). The proteins were then transferred onto a polyvinylidene difluoride membrane (Bio-Rad Laboratories, Hercules, CA, USA). The membrane was incubated overnight with primary antibodies of anti-CD9 (1:750), CD63 (1:500), CD81 (1:700), a tumor susceptibility gene (TSG) 101 (1:1000), and calnexin (1:500) (ab133615, Abcam, Cambridge, UK). After the membrane was washed, it was incubated for 2 h with secondary antibodies of goat anti-rabbit IgG (H+L) HRP (XD344763, ThermoFisher Scientific, Waltham, MA, USA) (1:5000) for calnexin, and goat anti-rabbit IgG HRP (SC2004, Santa Cruz Biotechnology, Dallas, TX, USA) (1:2000) for other targets. The blots were visualized with an ECL detection reagent (RPN2232, Cytiva, Marlborough, MA, USA), and read with a ChemiDoc^TM^ MP Imaging System (Bio-Rad Laboratories, Hercules, CA, USA). Protein bands were photographed and quantified in ImageJ.

### 2.5. Dynamic Light Scattering (DLS)

Eighty-one EXO pellets from 28 burn patients (23M/5F, 42.4 ± 4.1 y) and 13 controls (6M/7F, 36.1 ± 4.5 y) were suspended in 0.5 mL PBS (Ph = 7.4) by a gentle vortex. The sample was then diluted in PBS (10 μL sample in 0.75 mL PBS), and the EXOs in the samples were characterized by DLS by utilizing Zetasizer Nano ZS (Malvern Panalytical Ltd., Malvern, UK). The refractive index of EXOs was set as 1.39 and the absorption coefficient was set to 0.01.

### 2.6. Multiplex Cytokine Assay

Forty-one EXO pellets from 13 burn patients (10M/3F, 42.09 ± 3.87 y) and 6 controls (2M/4F, 36.08 ± 4.46 y) were lysed with RIPA buffer (20-188, MilliporeSigma, Burlington, MA, USA). The protein concentration was determined by employing BCA protein assay.

The EXO protein samples were diluted with PBS to 2–5 mg/mL, and 110 μL of each protein sample was utilized for multiplexed quantification by employing Luminex xMAP technology, including transforming growth factor beta (TGF-β) 3-plex discovery assay and inflammatory cytokine 15-plex discovery assay (MilliporeSigma, Burlington, MA, USA). The multiplex analysis was performed using the Luminex™ 200 system (Luminex, Austin, TX, USA).

### 2.7. EXO Uptake Assay

EXOs were dyed utilizing ExoGlow-protein EV green labeling reagent (EXOGP300A-1, System Biosciences, Palo Alto, CA, USA) according to the manufacturer’s instructions. Briefly, 300 μg of the EXOs were suspended in 0.5 mL PBS. Next, 500xlabeling dye (1 μL) was added to the EXO solution, vortexed, and incubated at 37 °C with shaking (350 rpm) for 20 min. ExoQuick-TC (167 μL) was added to the solution and incubated at 4 °C overnight. The stained EXOs were harvested by spinning them at 10,000 rpm for 10 min, and then the EXO pellet was resuspended in 100 μL PBS (3 μg/μL).

Human dermal fibroblasts explanted from human skin biopsies, as described previously [7], were plated on coverslips in the wells of a 12-well plate and incubated with the dyed EXOs (30 μg/mL) or the same volume of PBS for 24 h in FBS-free Dulbecco’s Modified Eagle’s Medium (DMEM).

The coverslips were washed with PBS and the cells were fixed with methanol before they were stained for alpha smooth muscle actin (⍺SMA), utilizing anti-⍺SMA rabbit monoclonal antibody (MABT381, MilliporeSigma, Burlington, MA, USA) and Alexa Fluor 546 goat anti-rabbit IgG (H=L) (A11010, ThermoFisher Scientific, Waltham, MA, USA). The coverslips were mounted with Prolong Gold antifade with DAPI (blue) (P36935, ThermoFisher Scientific, Waltham, MA, USA), and finally sealed on a glass slide. EXO uptake was observed under a fluorescent microscope (Eclipse Ti, Nikon Canada. Mississauga, ON, Canada).

### 2.8. Fibroblast Regulation by EXOs In Vitro

EXOs from the burn patients and controls were suspended in 0.5 mL PBS. The serum EXO concentration was determined utilizing BCA Protein Assay.

Three pairs of HTS and normal skin (NS) primary fibroblasts were isolated and cultured from human HTS and the site-matched NS tissues of burn patients, as previously described [8]. After they were starved in DMEM with 2% FBS overnight in the wells of 12-well plates, the NS fibroblasts were treated with EXOs (30 μg/mL) in FBS-free DMEM for 48 h. The NS and HTS cells were treated with the same volume of PBS as the controls.

Cell proliferation was determined by cell counting. The treated cells were detached with Gibco^TM^ TrypLE^TM^ express enzyme (12606028, ThermoFisher Scientific, Waltham, MA, USA) and washed with PBS. The live cells were counted per ml of cell suspension by a TC20^TM^ automated cell counter (Bio-Rad Laboratories, Hercules, CA, USA).

Cell apoptosis was investigated by staining for caspase-3/7. The treated cells were transferred to FACS tubes and suspended with 1 mL staining buffer. The cells were incubated in complete DMEM with 1 µL caspase-3/7 green detection reagent (CellEvent caspase-3/7 green flow cytometry assay kit, C10427, Life Technologies, Carlsbad, CA, USA) at room temperature for 60 min, protected from light. During the final 5 min, 1 μL of dead cell stain solution (1 mM) was added and mixed gently. The cells were analyzed without washing or fixing on Attune NxT utilizing a 488-nm laser, and the fluorescence emission was detected employing a 530/30 BP filter for the caspase 3/7 green stain and a 690/50BP filter for dead cell stain, respectively. Double positive cells were necrotic cells, and caspase 3/7 green positive cells were apoptotic. Normal and HTS fibroblasts were treated with PBS as controls.

### 2.9. Statistical Analysis

The patient numbers are presented as mean ± SD, and other data are displayed as mean ± SE. The groups were statistically compared utilizing one-way ANOVA with Post Hoc testing using SPSS. A *p*-value of ≤0.05 was considered statistically different. Linear regression was performed in the patient groups, organized by burn severity (TBSA) or time after burn injury. The linear coefficient R refers to a linear trend between two variables. The coefficient of variation or R^2^ value of more than 0.5 is considered meaningful.

## 3. Results

### 3.1. Morphology and Density of Serum EXOs

Patient EXOs were small with fuzzy edges at 2 days, larger but with lower density at 7 days, and multiple sizes with clear edges and lowest density at 63 days post-burn injury. Control EXOs were multiple sizes with clear edges (Figure 1A). The average EXO density was lower in the patients than in controls. In the patients, EXO density significantly decreased immediately after the burn injury and then increased over time during wound healing; the overall trend was upward with variability (Figure 1B).

### 3.2. Marker Expression of EXOs

EXOs both from patients and controls expressed CD9, CD63, CD81, and TSG101, but not calnexin (Figure 2A). The expression of the markers was low at the early stage of wound healing and then increased over time as wound healing progressed. Fibroblast lysates from human dermal fibroblasts expressed TSG101 at a low intensity and calnexin at a strong intensity (Figure 2B).

### 3.3. Concentration of EXOs

The EXO concentration in the serum was lower for the patients with ≥20% TBSA than for the controls and patients with <20% TBSA. In the patients, the EXO concentration was negatively correlated with TBSA and significantly lower at 0–14 days than at ≥64 days post-burn injury (Figure 3A). The EXOs contained a lower level of proteins in the patients with ≥20% TBSA and at <14 days post-burn injury than in controls. The protein concentration of EXOs was negatively correlated with TBSA in the patients (Figure 3B).

### 3.4. EXO Size Distribution

DLS exhibited a monomodal profile in most EXO samples. The representative images show that EXO size distribution decreased in intensity and increased in peak over time during wound healing in one patient (M, 45 y, 15 % TBSA). Two controls (1F/1M, 41/52 y) had a similar EXO size distribution (Figure 4A).

EXO size distribution changed according to distinctive patterns in the patients. Compared to continuous multiple-size EXOs in the controls, patients with <20% TBSA had 4 small populations of EXOs with multiple sizes at 0–7 days, 2 populations with a wide range of larger sizes at 8–14 days, 1 large population with larger size at 15–21 days, 1 large population with the largest size at 22–28 days, 2 populations with decreased sizes at 29–63 days, and 1 population with a wide range of larger size after 64 days post-burn injury. The EXO size distribution started decreasing after 4 weeks post-burn injury. The EXO size distribution was dependent on burn severity or TBSA in the patients. The higher the TBSA, the earlier the EXO size distribution began to change, starting in the first week for the patients with 40–59% and ≥60% TBSA; in the second week for the patients with 20–39% TBSA; and the third week for the patients with <20% TBSA post-burn injury (Figure 4B).The EXOs exhibited multiple sizes in 0–7 day open wounds; they were larger when the wounds were partially closed and largest after the wounds were completely closed (Figure 4C).

### 3.5. Cytokine and Cytokine Correlation with Patient TBSA

By measuring growth and inflammatory cytokines from lysed EXOs, we found that IL-6 and IL-8 were increased, but TGF-β3, interleukin (IL)-1β, IL-1RA, and IL-12p40 were decreased in all patients as compared to controls. TNF-α, interferon (IFN)-γ, monocyte chemoattractant protein (MCP)-1, and IL-12p70 were increased in the patients with <40% TBSA and decreased in the patients with ≥40% TBSA. TGF-β1, TGF-β2, IL-10, and IL-13 were not changed in the patients with <40% TBSA but decreased in the patients with ≥40% TBSA. IL-2, IL-4, IL-5, and GM-CSF were undetectable (Figure 5A). Linear regression was performed for each cytokine at varying burn severity or TBSA and at each post-burn sampling. The levels of IL-6, IL-8, TGF-β1, and TGF-β2 positively correlated with the TBSA before 29 days; IFN-γ, IL-1RA, and MCP-1 positively correlated with the TBSA at 15–21 days, but negatively correlated at 29–63 days; IL-12p40 positively correlated with TBSA at 15–21 days, but negatively correlated after 21 days; TGF-β3 positively correlated with TBSA at 22–28 days, but negatively correlated after 28 days; IL-1β, IL-10, IL-12p70, IL-13, and the tumor necrosis factor (TNF)-α negatively correlated with TBSA after 21 days (Figure 5B).

### 3.6. EXO Uptake by Dermal Fibroblasts

Dermal fibroblasts expressed ⍺SMA (red) and took up dyed EXOs (green), as the arrow indicates in Figure 6.

### 3.7. Fibroblast Regulation by EXOs

HTS fibroblasts proliferated more quickly but had more apoptotic cells than the NS fibroblasts. EXOs both from patients and controls stimulated cell proliferation and apoptosis in NS fibroblasts without significant differences between patients and controls (Figure 7).

## 4. Discussion

It has been 40 years since EXOs were discovered in immature red blood cells in 1983 [9,10] and later named in 1987 [11,12]. EXOs are lipid bilayer membrane-bound vesicles with a size range of 40~160 nm in diameter, containing proteins, lipids, nucleic acids, and metabolites from cells of origin. Nucleic acids include DNA, messenger RNA, microRNA, and many other non-coding RNAs [4,13]. EXOs are thought to transfer biological information between cells and therefore contribute to the physiological process of intercellular communication, tissue repair, regeneration and rejuvenation, waste management, and immune response [4,14,15]. EXOs were also found to be involved in infection, metabolic and cardiovascular diseases, neurodegenerative disorders, cancer, inflammation, and coagulation [4,13,15,16,17,18]. Recently, many translational medical studies of EXOs as diagnostic tools and treatment approaches have elicited widespread attention [4,5,19,20]. For cutaneous wound healing specifically, mesenchymal cells (MSCs), adipose stem cells (SCs), and umbilical cord blood-derived EXOs have shown promising effects in regulating inflammation, fibroblast activation, collagen production, neovascularization, and regeneration [18,21,22,23,24,25,26,27]. EXOs have been reported to ameliorate fibrosis in the heart, kidneys, liver, lungs, and skin [23,28,29].

In this study, serum EXOs were found to respond to burn injury with decreases in size, density, and protein concentration. Dynamic changes over time after the initial injury included increases in variability during wound healing, some of which were also dependent on burn severity. The changes in EXO features may be involved in the quality of wound healing and the development of fibrosis after injury, suggesting the clinical significance of EXOs in burn care.

EXOs contain proteins associated with secreting cells, and some of these proteins are unique to cellular homeostasis by participating in cell adhesion, structural dynamics, membrane fusion, metabolism, and signal transduction [13,17]. Surface proteins of CD9, CD63, and CD81 are involved in cell targeting and adhesion. TSG101 is involved in multivesicular body biogenesis, while heat shock proteins (HSP)-70 and HSP-90 are molecular chaperones [30]. In this study, the changes in EXO concentration in the serum and protein concentration of lysed EXOs were consistent with the changes in the EXO density, as demonstrated by counting the EXO number in 5 TEM images in the patients. The serum EXOs isolated both from burn patients and healthy individuals were found to express common markers of CD9, CD63, CD81, and TSG101. HSP-70 and an internal control protein of β-actin were undetectable both in patient and control EXOs, which may be associated with the low protein expression or cellular origin.

EXOs also contain cytokines, transcription factor receptors, growth factor receptors, and other bioactive molecules [13]. Cytokines are vital participants in wound healing by transferring signals to local cells in the wound. Cytokines present at the right time and number may enhance the immune response, control inflammation, and promote tissue repair. However, cytokine overproduction often leads to increased inflammation or proliferation commonly seen in scar formation [31]. Some growth and inflammatory cytokines were detected in the EXOs from burn patients. Although the levels of TGF-β1 and TGF-β2 were low in patients with serious burns, they increased with increasing burn severity in the proliferation phase of wound healing. TGF-β3 decreased in all patients, especially in those with larger TBSA or more severe burns despite proliferating at the early remodeling phase. TGF-β released by platelets and macrophages mediates fibroblast migration to the wound site to trigger normal wound healing in the proliferation phase. During re-epithelization, TGF-β produced by M2 macrophages and keratinocytes induces angiogenesis, epithelial cell proliferation, and cell migration bordering the wound edges [32]. TGF-β has three isotypes, TGF-β1, TGF-β2, and TGF-β3, where TGF-β1 and TGF-β2 are thought to be involved in fibrotic cutaneous wound healing, but TGF-β3 has been noticed because of its antifibrotic effects [33].

Pro-inflammatory cytokines are essential for inflammation in wound healing. Anti-inflammatory cytokines, in contrast, promote proliferation and remodeling. A balance of inflammatory cytokines is key to facilitating normal wound healing. Prolonged inflammation caused by pro-inflammatory cytokines or overproduction such as that of IL-1, IL-6, IL-8, and TNF-a may lead to fibrotic wound healing [3,31]. After burn injury, the levels of IL-6 and IL-8 elevated in all patients, unlike in healthy individuals, and they were trending upward with increasing burn severity. They are known to contribute to wound healing by enhancing immune response but may be involved in prolonged inflammation and fibrotic wound healing in serious burns [30]. Other inflammatory cytokines, IL-1β, IL-1RA, Il-10, IL-12p70, IL-13, TNF-a, and IFN-γ not only changed in levels after burn injury but also were trended downward when the burn severity increased in the remodeling phase. IL-10 and IFN-γ, as anti-inflammatory cytokines, were reported to promote regenerative healing and prevent scar formation by in vivo and in vitro experiments [34,35].

EXO size distribution changed, following distinct patterns in burn patients, which was dependent on burn severity and time after burn injury and in the stages of healing. EXO heterogeneity in size and content reflects different signals carried from the cell origin and impacts recipient cells differently [4]. Although regulation of some growth and inflammatory cytokines contained by serum EXOs was seen after burn injury, further proteomic and RNA sequencing analysis may provide more signal information carried by the EXOs. By correlating more clinical information on the patients who form scars and their severity with changes in serum EXOs characteristics, breakthroughs in the understanding of the role of EXOs in the serum in wound healing and scar formation will help to develop methods to prevent and treat scarring after burn injury.

To examine the role of EXOs in fibroblast regulation, EXO uptake by human dermal fibroblasts was first confirmed after the cells were incubated with dyed EXOs. EXO uptake by recipient cells may occur after interferon gamma receptor 1-ligand IFN-γ interactions or by direct membrane fusion and endocytosis [13]; however, further details regarding the mechanism by which serum EXOs uptake by dermal fibroblasts will require further investigation.

EXOs both from burn patients and healthy individuals stimulated cell proliferation and apoptosis without significant differences between patients and controls despite varying TBSA groups and time after burn injury. We thought it may be caused by limited exosome samples in each group and rough grouping by burn area (TBSA). Our study has shown that the inflammatory cytokine levels that exosomes contained were correlated both to burn area and the wound-healing stage. In further study, an increase in patient number and data analysis by more precise grouping by wound healing time have been considered. Similar observations were made in colorectal cancer [36], where EXOs derived from colorectal cancer fibroblasts in vitro promoted proliferation, apoptosis, migration, invasion, and angiogenesis of colorectal cells. The fibroblast regulation in cell proliferation and apoptosis may be in response to another yet unidentified factor. Furthermore, the response of dermal cells to EXOs may vary in wound microenvironments *in vivo*.

## 5. Conclusions

EXOs responded to burn injury through a change in morphology, density, size, protein concentration, and specific cytokine expression. All these changes were correlated with burn severity and time of wound healing. Combining clinical features of wound healing in burn patients suggests a possible or important role of EXOs in wound healing and scarring as well. Further study of the mechanism will be considered. EXOs both from burn patients and healthy individuals stimulated cell proliferation and apoptosis *in vitro.* However, our study is limited by the number of patient samples available, particularly of burns of greater severity. Further understanding will require more patients and more time points post-burn injury to understand the effect of serum EXOs on dermal wound healing.

## Figures and Tables

**Figure 1 cells-12-01738-f001:**
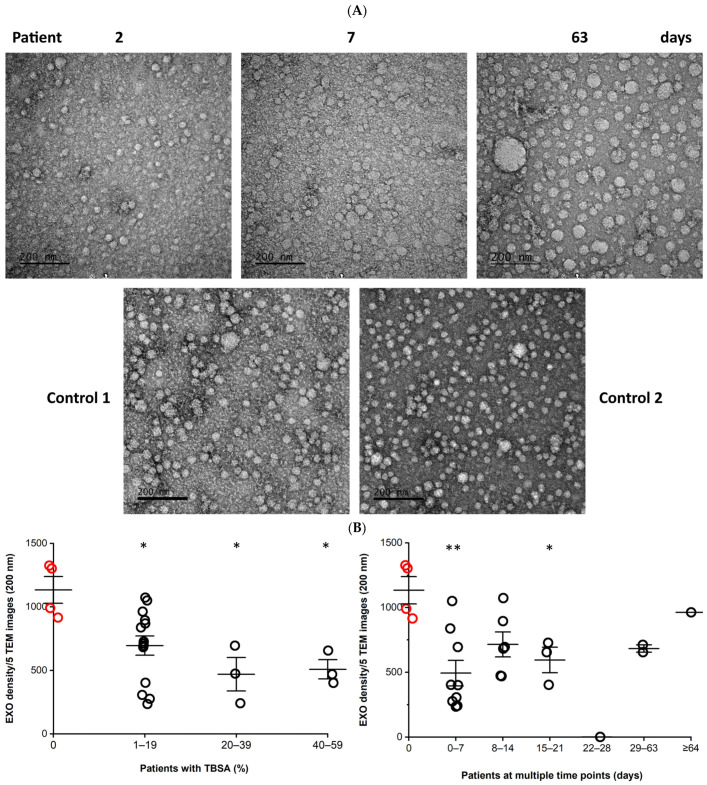
**Morphology and density of EXOs.** Each EXO pellet isolated from 1 mL serum of burn patients and controls was suspended in 0.5 mL PBS. (**A**) EXO morphology was observed under a transmission electron microscope (TEM) at 100 K magnification. Representative images show the EXOs from a patient with 15% TBSA (M, 45 y) at three points post-burn injury and two controls (M, 52 y and M, 24 y). Scale bar = 200 nm. (**B**) EXO density was determined by counting the EXO number in 5 TEM images from each sample. Data are displayed as mean ± SE. Patients (black circles), n = 6, 4M/2F, 48.0 ± 7.3 y; Controls (red circles), n = 4, 3M/1F, 30.9 ± 14.3 y. Patients vs controls, *, *p* ≤ 0.05; **, *p* ≤ 0.01.

**Figure 2 cells-12-01738-f002:**
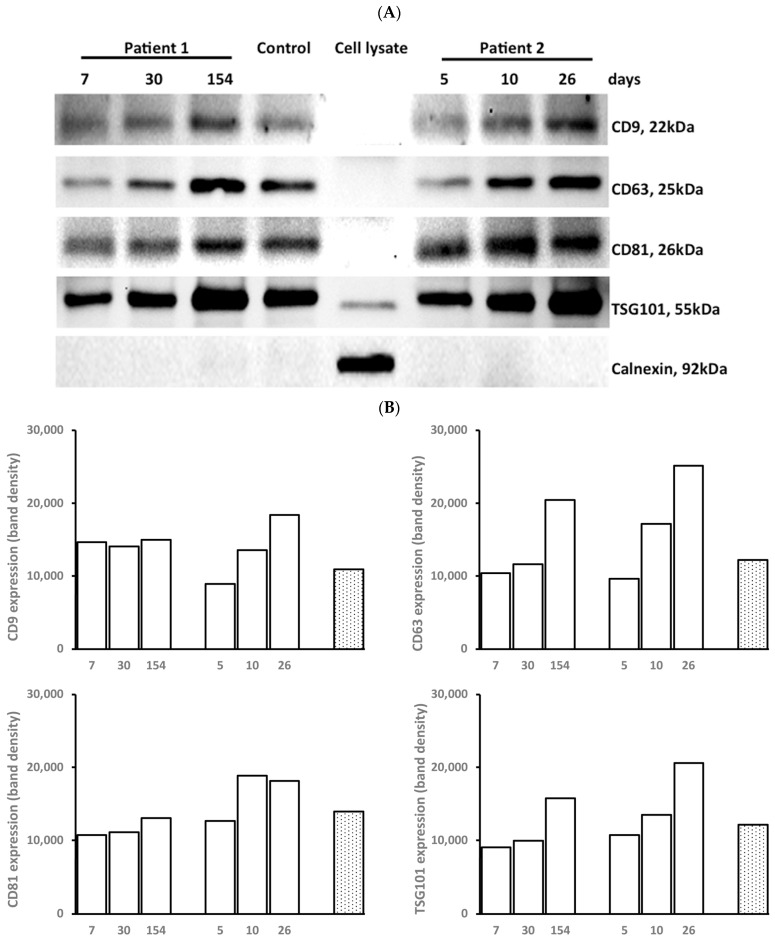
**Marker expression of EXOs.** Each EXO pellet isolated from 1 mL serum from patients and controls was suspended with 0.5 mL PBS. The concentration was determined using BCA protein assay. Marker expression of EXOs by Western blot. (**A**) The images show the marker expressions of CD9, CD63, CD81, and TSG101 in EXOs from two burn patients at three post-burn intervals (M, 50% TBSA, 53.6 y and M, 30% TBSA, 47.8 y) and a control (F, 56.6 y). (**B**) The density of bands quantified from the above images.

**Figure 3 cells-12-01738-f003:**
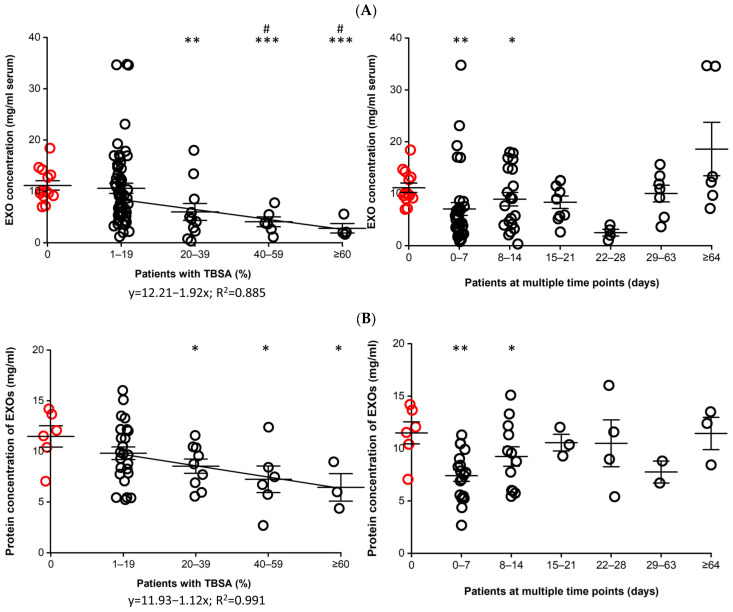
**Concentration of EXOs.** (**A**) EXO concentration in the serum. Each EXO pellet isolated from 1 mL serum of patients and controls was suspended with 0.5 mL PBS. The concentration was determined using BCA protein assay. Data are displayed as mean ± SE (mg/mL serum). Patients, n = 29, 23M/6F, 45.0 ± 3.2 y; Controls, n = 13, 6M/7F, 36.1 ± 4.5 y. Patients vs controls, *, *p* ≤ 0.05; **, *p* ≤ 0.01; ***, *p* ≤ 0.001. Patients with ≥20% TBSA vs. patients with <20% TBSA, #, *p* ≤ 0.05. (**B**) Protein concentration of EXOs. Each EXO pellet isolated from 1 mL serum of patients and controls was lysed with 0.5 mL RIPA lysis buffer. Protein concentration was measured using a BCA protein assay. Data are displayed as mean ± SE (mg/mL). Patients (black circles), n = 13, 10 M/3 F, 42.1 ± 3.9 y. Controls (red circles), n = 6, 2 M/4 F, 36.1 ± 4.5 y. Patients vs controls, *, *p* ≤ 0.05; **, *p* ≤ 0.01.

**Figure 4 cells-12-01738-f004:**
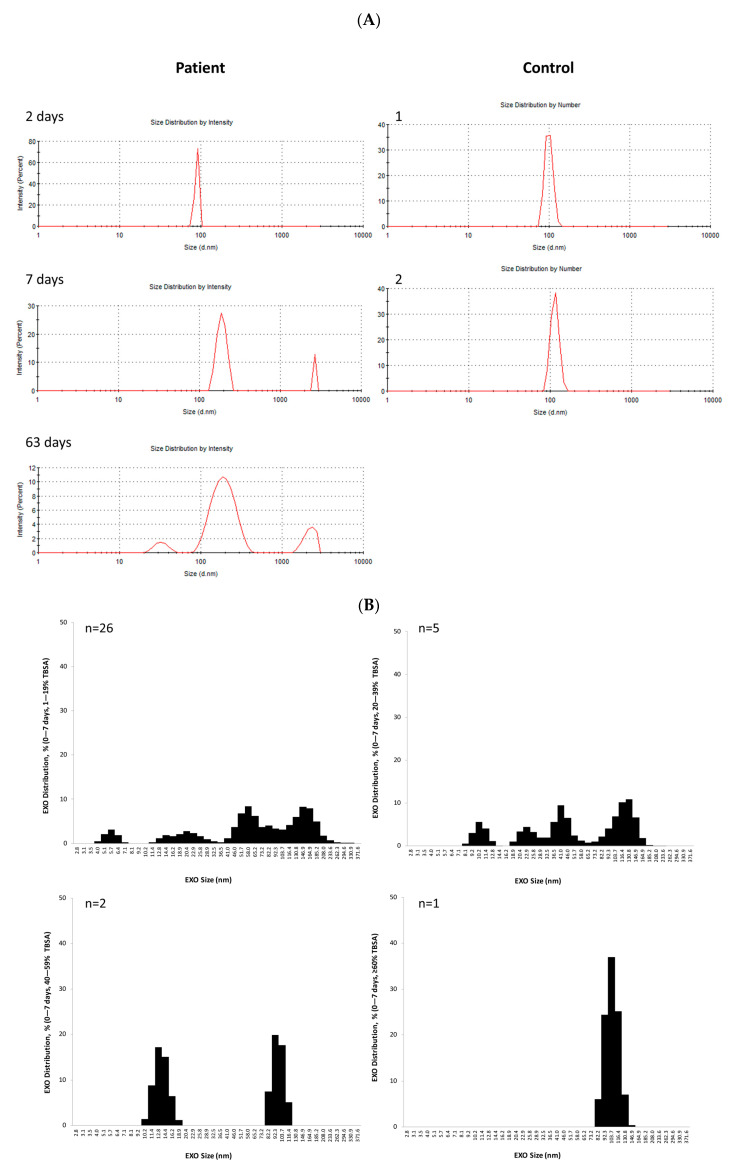
**EXO size distributions.** Each EXO pellet isolated from 1 mL serum was suspended by 0.5 mL PBS. A total of 10 µL of the EXO solution was diluted in 0.75 mL of PBS, and then the EXO size was analyzed by dynamic light scattering (DLS). The refractive index was set as 1.39, and the absorption coefficient was set to 0.01. (**A**) shows representative DLS images of serum EXOs from a patient with 15% TBSA at 3 intervals post-burn injury (M, 45 y) and 2 controls (F, 41 y and M, 52 y). (**B**) EXO size distributions are grouped by TBSA and time points post-burn injury. Patients, n = 28, 23 M/5 F, 42.4 ± 4.1 y; Controls, n = 13, 6 M/7 F, 36.1 ± 4.5 y. (**C**) EXO size distributions are grouped by wound status: 0–7 days open wound, half closure, and closure. Patients, n = 28, 23M/5F, 42.4 ± 4.1 y; Controls, n = 13, 6M/7F, 36.1 ± 4.5 y.

**Figure 5 cells-12-01738-f005:**
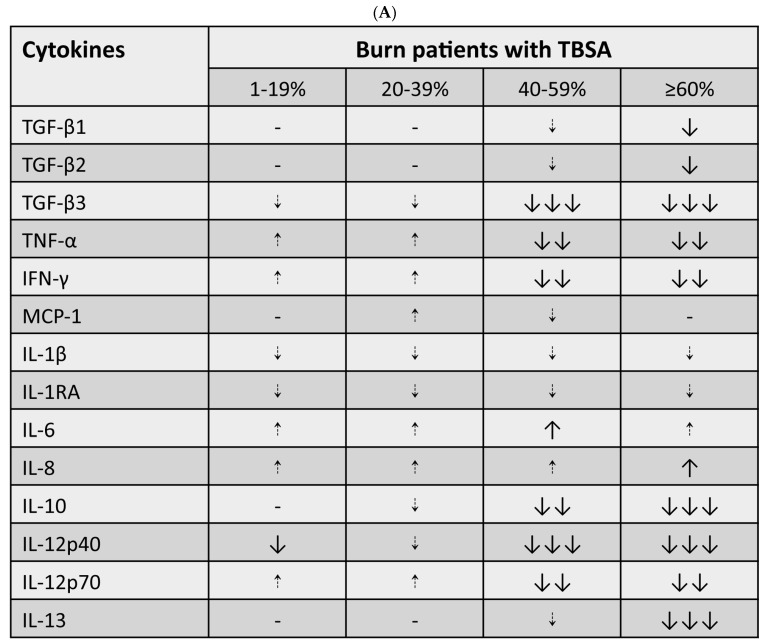
**Cytokine and cytokine correlation to patient TBSA.** Each EXO pellet from burn patients and controls was lysed with RIPA lysis buffer. Protein concentration was determined using the BCA protein assay. After the protein samples with PBS were adjusted to a concentration of 2–5 mg/mL, 110 µL of protein sample was run for multiplex cytokine quantification utilizing Luminex xMAP technology. (**A**) Cytokine changes relative to controls in burn patients. ↑ or ⇡, increase; ↓ or ⇣, decrease; dotted arrow, no statistical significance; solid arrow, statistical significance; 1 arrow, *p* ≤ 0.05; 2 arrows, *p* ≤ 0.01; 3 arrows, *p* ≤ 0.001; -, no change. (**B**) Line regressions were performed between each cytokine at each post-burn sampling and TBSA. A value of R^2^ of more than 0.5 is considered meaningful. The graph shows cytokine correlations with TBSA at various post-burn sampling points of the patients. Patients, n = 13, 10M/3F, 42.1 ± 3.9 y; Controls, n = 6, 2M/4F, 36.1 ± 4.5 y.

**Figure 6 cells-12-01738-f006:**
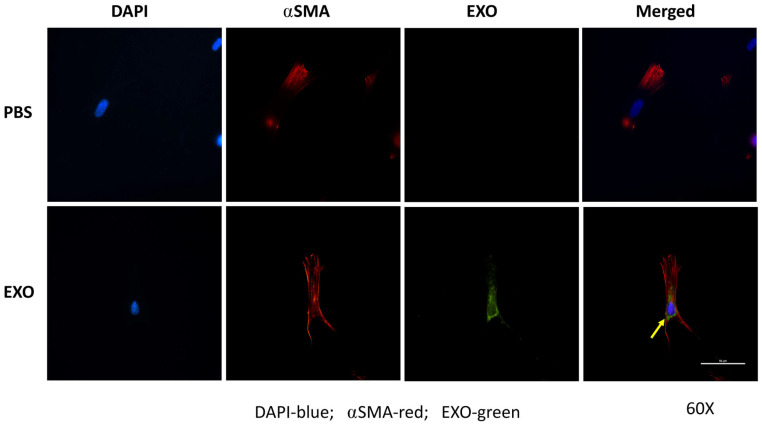
**EXO uptake by dermal fibroblasts.** Dermal fibroblasts were plated on a coverslip and incubated with dyed EXOs (30 μg/mL) or PBS (control) for 24 h in FBS-free DMEM. After the cells were washed with PBS, they were fixed with methanol and stained for ⍺SMA. The coverslip was mounted with Prolong Gold antifade with DAPI and sealed on a glass slide. EXO uptake was finally observed under a fluorescent microscope.

**Figure 7 cells-12-01738-f007:**
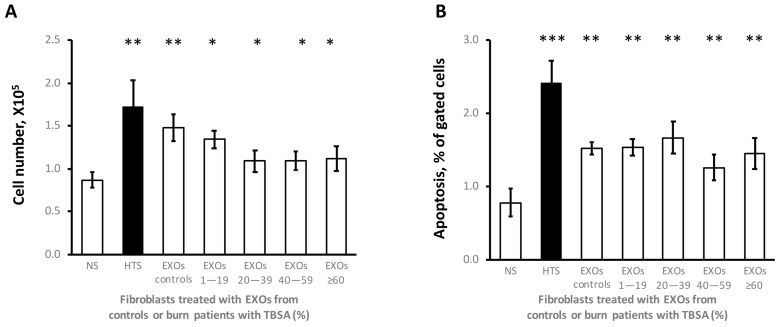
**Fibroblast functional regulation by EXOs.** Three pairs of hypertrophic scar (HTS) and site-matched normal skin (NS) fibroblasts were cultured to 80% confluence in 12-well plates. After the NS fibroblasts were starved in the DMEM with 2% FBS overnight, they were treated with EXOs (30 μg/mL) in FBS-free DMEM for 48 h. NS and HTS fibroblasts were treated with PBS as controls. (**A**) Cell proliferation was determined by counting cells. (**B**) Apoptosis was assessed by caspase 3/7-expressing cells. Patient EXOs, n = 31 from 19 patients (14M/5F, 42.7 ± 3.4 y); Control EXOs, n = 6 from 6 controls (2M/4F, 43.3 ± 7.4 y). NS treated with EXOs vs. NS treated with PBS, *, *p* ≤ 0.05; **, *p* ≤ 0.01, ***, *p* ≤ 0.001.

## Data Availability

We would like to share research data on request.

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
