# Peer review of "Characteristics of Serum Exosomes after Burn Injury and Dermal Fibroblast Regulation by Exosomes In Vitro"

_cells, 2023, doi:10.3390/cells12131738_

Round 1
Reviewer 1 Report
Dear authors
I have gone through your manuscript and found it quite interesting, where you embarked on the activity of exosomes derived from patients and their effectiveness in treating burn wounds. I have some suggestions and reservations about your paper listed below.
1. In the abstract in the very first line, you have written the exosomes mechanism of wound healing, especially burn, but in your manuscript, it did not have any such study e.g., investigating their effectiveness in treating wounds in human volunteers, or cell line (scratch test). Therefore, it is advised to be careful in writing scientific papers.
2. The introduction lacks exosomes already used for investigating their effectiveness in skin regeneration following injury, as hundreds of such articles are available e.g., just look at a review article (https://www.ncbi.nlm.nih.gov/pmc/articles/PMC10086759.). Therefore, incorporate the necessary literature to support your claim.
3. Total exosomes isolation from serum have repetition of a sentence i.e., centrifugation of serum. Please correct it.
4. Similarly, in TEmM, its Hepes buffer solution, not Hepes solution. Correct it.
5. The TEM images lacks magnification used for observing the exosomes. Please add.
6. Figure 2C bars axis are merely visible. Please enhance figure axis resolution. The same goes for figure 3.
The English language needs minor editing, especially grammatical. e.g., a sentence never starts with a number; for instance look at section 2.1, line 3, it shall be thirteen instead of 13.
Author Response
Dear Reviewer,
Thank you very much for reviewing our manuscript. We revised our manuscript based on your valuable suggestions below:
- In the abstract in the very first line, you have written the exosomes mechanism of wound healing, especially burn, but in your manuscript, it did not have any such study e.g., investigating their effectiveness in treating wounds in human volunteers, or cell line (scratch test). Therefore, it is advised to be careful in writing scientific papers.
Many studies reported that cell-origin exosomes promoted wound healing without detail mechanisms. We aim to explore how the exosomes play the role and if the exosome behaviour relates to scarring. To investigate the exosome behaviour post burn injury is a beginning of our goal in this field.
- The introduction lacks exosomes already used for investigating their effectiveness in skin regeneration following injury, as hundreds of such articles are available e.g., just look at a review article (https://www.ncbi.nlm.nih.gov/pmc/articles/PMC10086759.). Therefore, incorporate the necessary literature to support your claim.
We have referred the paper.
- Guang Yang, Saquib Waheed, Cong Wang, Mehdihasan Shekh, Zhibin Li, Jun Wu. Exosomes and Their Bioengineering Strategies in the Cutaneous Wound Healing and Related Complications: Current Knowledge and Future Perspectives. International Journal of Biological Sciences 2023; 19(5): 1430-1454. doi: 10.7150/ijbs.80430
Reference 4, 5, 6 cited in the section of Introduction are also kinds of review papers in the past 3 years about the exosome therapeutic effects of wound healing.
The reference order changed.
- Total exosomes isolation from serum have repetition of a sentence i.e., centrifugation of serum. Please correct it.
We have corrected.
- Similarly, in TEmM, its Hepes buffer solution, not Hepes solution. Correct it.
Corrected it.
- The TEM images lacks magnification used for observing the exosomes. Please add.
Added the magnification in Figure 1 legend for the TEM images.
- Figure 2C bars axis are merely visible. Please enhance figure axis resolution. The same goes for figure 3.
Improved the figure axis resolution by Increasing resolution and figure size in Figure 2C (change to Figure 2C and 2D) and Figure 3 A, 3B, 3C. Figure 2 legend and section 3.2 were modified too.
Comments on the Quality of English Language
The English language needs minor editing, especially grammatical. e.g., a sentence never starts with a number; for instance look at section 2.1, line 3, it shall be thirteen instead of 13.
Checked all text and corrected them. Thank you.
Reviewer 2 Report
The authors present a longitudinal observational study to demonstrate changes in the presence of EXOs in serum isolated from adult patients with burn injuries. It is an interesting study albeit with limited number of patients and further limited by detailed study of even smaller sample sizes for some of the experiments. Whilst this study demonstrates the presence of the EXOS in the serum and some changes with TBSA and over time course post-injury, it is not feasible to categorically tie up the cause and relationship of the findings to the outcome of burn injury or indeed to the post-burn scarring. The authors acknowledge these deficits and the need for further work. However, as a concept, this study does contribute to the knowledge base and lays a foundation for future course of work required.
Author Response
Thank you so much.
Reviewer 3 Report
The article is well-written and provides significant results regarding the changes in exosome profiles in burn victims as the injury is healing. The results are well-explained and easy to follow. This is interesting data and it is a step forward towards a better understanding of the role of exosomes in wound healing.
1) However, the application of exosomes from both burn victims and healthy individuals had the same effect on fibroblast. I consider this to be a key point of the article. It would be helpful to explain more about these results. How does this relate to future research? Why did the fibroblast reacted the same, even though the exosome content was significantly different between the sample?
2) In the results section, please separate the results referring to the validation of isolated vesicles as being exosomes (such as CD63, CD81, CD9, or TEM size evaluation) from the results related to the difference in exosomes characteristics between controls and burn victims ( changes in size, protein content and cytokines in exosomes )
3) In the conclusion it should be explained more how are the results presented in the article relevant for the further study of regenerative medicine.
Author Response
Dear Reviewer,
Thank you very much for reviewing our manuscript. We revised it based on your valuable suggestions below:
1) However, the application of exosomes from both burn victims and healthy individuals had the same effect on fibroblast. I consider this to be a key point of the article. It would be helpful to explain more about these results. How does this relate to future research? Why did the fibroblast reacted the same, even though the exosome content was significantly different between the sample?
There was no different regulations of fibroblast growth and apoptosis by exosomes from both burn patients and healthy individuals. We thought it may be caused by limited exosome samples in each group and rough grouping by burn area (TBSA). Our study has shown that the inflammatory cytokine levels that exosomes contained were correlated to both of burn area and wound healing stage. In further study, an increase in patient number and data analysis by more precise grouping by wound healing time have been considered. It was added to the section of Discussion.
2) In the results section, please separate the results referring to the validation of isolated vesicles as being exosomes (such as CD63, CD81, CD9, or TEM size evaluation) from the results related to the difference in exosomes characteristics between controls and burn victims ( changes in size, protein content and cytokines in exosomes )
Reordered the figures:
Old Figure 2C changed to Figure 2A and 2B.
Old Figure 2A and AB changed to Figure 3A and 3B.
The text related to Figure 2 and 3 was adjusted too.
3) In the conclusion it should be explained more how are the results presented in the article relevant for the further study of regenerative medicine.
Combining clinical features of wound healing in burn patients, the exosome behaviour post burn injury suggests a possible or important role of EXOs in wound healing and scarring as well. Further study in the mechanism will be considered. It was added in the conclusion.